# LANGUAGE MODEL MERGING IN ITERATIVE PREFERENCE LEARNING

## ABSTRACT

Learning from preferences has become a scalable paradigm for training high-capacity language models, as it is not limited to human-produced data, allowing models to surpass human performance. Advanced feedback learning algorithm is typically online or iterative for high sample efficiency. Among these, iterative preference optimization is popular due to its simplicity, efficiency, and robustness. However, in iterative preference optimization, models do not necessarily achieve optimal performance since they sequentially learn data from with different distributions. A simple way to bridge the gap is model ensemble, which incurs excessive inference costs. Inspired by the theoretical analysis for preference learning, we propose a simple model merging strategy that approximates model ensemble without additional training and inference costs, leading to Pareto-superior models.

## 1 INTRODUCTION

Large language models have acquired strong foundational capabilities and zero-shot performance on held-out tasks thanks to language modeling (Radford et al., 2018; Devlin et al., 2019) and instruction tuning (Wei et al., 2022a; Sanh et al., 2022). Despite these advancements, language models do not naturally assign high probability on human-favorable reliable, safe, and helpful responses. Meanwhile, as high-quality text has been exhaustively crawled and language models have approached human-level on many tasks, it is difficult for models to further learn from supervised learning (Touvron et al., 2023; Burns et al., 2024). An emerging diagram is reinforcement learning from (human) feedback (Christiano et al., 2017; Ouyang et al., 2022), which does not rely on human to produce gold label but learns from human satisfaction or environment feedback, allowing the model to surpass human-level performance.

Traditionally, the policy is optimized by standard reinforcement learning algorithms such as proximal policy optimization (PPO; Schulman et al., 2017). Unfortunately, despite its strong performance (Xu et al., 2024; Ivison et al., 2024), PPO is notorious for being resource-demanding. This is mainly caused by two factors: (i) PPO requires to generate completions from the present policy. However, to fit large language models to limited per-device memory, they are typically distributibly trained, which suffers low throughput for auto-regressive generation (Touvron et al., 2023; Hu et al., 2024); (ii) PPO requires four models: policy model, reference model, reward model, and value model, to be loaded simultaneously, which is memory intensive (Li et al., 2024; Shao et al., 2024).

Recently, many economical alternatives have been proposed (Wu et al., 2024; Meng et al., 2024), among which the most popular one is direct policy optimization (DPO; Rafailov et al., 2023). The design of DPO is based on the observation that each policy induces a reward model, where the policy is optimal under the reward model. To optimize the policy, it suffices to train the reward model. DPO is initially conducted on pre-collected fixed preference datasets, which struggle to cover the board space of natural language (Dong et al., 2024; Zhang et al., 2024) and can bring regression (Guo et al., 2024). To bridge the gap, preference learning is implemented in an iterative way, where in each iteration, completions are sampled from the latest checkpoint, annotated by the verifier, and preference learning is conducted to produce the next checkpoint, leading to consistent improvement (Dong et al., 2024). Iterative preference learning demonstrates remarkable performance and has been applied to align the state-of-the-art open-weight models, such as Llama (Dubey et al., 2024) and Qwen (Yang et al., 2024).

In iterative DPO, the reward model sequentially learns preference data sampled from different policies, which are not independent and identically distributed as in the standard supervised learning (Vapnik, 2013). We hypothesize that such training may deprive the performance of the final reward model and policy. A simple way to mitigate the loss is through model ensemble, which blends the token distributions of multiple language models at inference time (Mitchell et al., 2024; Liu et al., 2024). However, this method incurs an inference cost that increases linearly with the number of models involved. Fortunately, inspired by the theoretical analysis of preference learning, weight averaging between the reference policy and iterative DPO checkpoints can approximate the ensemble model (section 3.1). On this basis, we propose a simple yet effective merging strategy, where two hyper-parameters control the magnitude and direction of the enumerated ensemble reward model, respectively (section 3.2). Despite the highly nonlinear nature of deep neural networks, the approximation works surprisingly well (section 4.1). We apply the merging strategy to the iterative preference learning of two advanced open-weight models, *i.e.*, Llama-3 (Dubey et al., 2024) and Qwen2 (Yang et al., 2024), leading to models with better foundational capabilities and alignment (section 4.2). We also observe that the merged models induce more accurate reward models, providing evidence for our hypothesis (section 4.3).

## 2 PRELIMINARY

Conventionally, preference learning consists of two steps, *i.e.*, reward modelling and policy optimization (Christiano et al., 2017; Ouyang et al., 2022). In reward modeling, a reward model $r_\phi$ is trained to fit the collected preferences. A common assumption is that the preferences are sampled from a Bradley-Terry model (Bradley & Terry, 1952). Denote $\mathcal{X}$ and $\mathcal{Y}$ as the prompt and completion set, respectively. For prompt $x \in \mathcal{X}$ and completion pair $(y_1, y_2) \in \mathcal{Y}^2$, the probability that $y_1$ is preferred over $y_2$ is

$$p(y_1 \succ y_2) = \sigma(r(x, y_1) - r(x, y_2)),$$

where $r : \mathcal{X} \times \mathcal{Y} \to \mathbb{R}$ is a latent reward function that we do not have access to and $\sigma$ is the sigmoid function. Now let $\mathcal{D} = \{(x, y_w, y_l)\}$ be the preference dataset, where $y_w$ and $y_l$ is the chosen and rejected completions, respectively. A reward model $r_\phi : \mathcal{X} \times \mathcal{Y} \to \mathbb{R}$ is trained to estimate the latent reward function $r$ following the loss

$$\mathcal{L}(\phi; \mathcal{D}) = -\mathbb{E}_{(x, y_w, y_l) \sim \mathcal{D}}[\log \sigma(r_\phi(x, y_w) - r_\phi(x, y_l))]. \tag{1}$$

The reward model $r_\phi$ serves as a proxy for the latent reward function $r$ in the subsequent policy optimization, whose objective is

$$\mathcal{J}(\pi; \mathcal{D}, r) = \mathbb{E}_{x \sim \mathcal{D}, y \sim \pi(\cdot|x)}[r(x, y)] - \beta \mathbb{E}_{x \sim \mathcal{D}}[D_{\mathrm{KL}}(\pi(\cdot|x)||\pi_{\mathrm{ref}}(\cdot|x)], \tag{2}$$

where $\pi_{\mathrm{ref}}$ is the reference policy, *e.g.*, the instruction-tuned model, and $\beta$ governs the weight of the KL regularization, preventing reward hacking (Stiennon et al., 2020).

DPO unifies the two steps by observing that eq. (2) can be solved analytically (Rafailov et al., 2023). Concretely, for any reward function $r$, the optimal solution $\pi^*$ to eq. (2) satisfies

$$\log \pi^*(y|x) = \log \pi_{\mathrm{ref}}(y|x) + \frac{r(x, y)}{\beta} + \mathrm{const}, \tag{3}$$

where the constant normalizes $\pi^*$ to be a policy, *i.e.*, $\sum_{y \in \mathcal{Y}} \pi^*(y|x) = 1$. Reversely, any policy $\pi$ induces a reward

$$r(x, y) = \beta \log \pi(y|x) - \beta \log \pi_{\mathrm{ref}}(y|x) \tag{4}$$

such that $\pi$ is the optimal solution to eq. (2) under the reward eq. (4). On this basis, to train a policy $\pi_\theta$ parameterized by parameter $\theta$, it suffices to fit the induced reward $r_\theta$ to the preference dataset. In the following context, we denote the reference policy as $\pi_{\theta_0}$ since it serves as the initialization of the parameter $\theta$. Substituting eq. (4) into eq. (1), we yield the DPO loss

$$\mathcal{L}(\theta; \mathcal{D}) = -\mathbb{E}_{(x, y_w, y_l) \sim \mathcal{D}} \left[ \log \sigma \left( \beta \log \frac{\pi_\theta(y_w|x)}{\pi_{\theta_0}(y_w|x)} - \beta \log \frac{\pi_\theta(y_l|x)}{\pi_{\theta_0}(y_l|x)} \right) \right].$$

For a better data coverage, DPO is conducted in iterative fashion, where the algorithm is illustrated in alg. 1. Rigorously, the newly collected data should be merged with data in previous iterations and the model should be initialized as the reference policy in each iteration (Bai et al., 2022), which leads to a computational complexity of $\mathcal{O}(T^2)$. For the consideration of efficiency, only the newly collected data is used for training and the model is initialized from the checkpoint in the last iteration. This is equivalent to training the model sequentially with the data collected from each iteration (Dong et al., 2024).

---

**Algorithm 1:** Iterative DPO

**Input:** number of iterations $T$, prompt sets $\mathcal{X}_{1:T}$, reference policy $\pi_{\theta_0}$, sampling budget $K$

**for** $t = 1, \ldots, T$ **do**

    Sample $y_{1:K} \sim \pi_{\theta_{t-1}}(\cdot|x), \forall x \in \mathcal{X}_t$

    $(x, y_w, y_l) \leftarrow$ label_pref$(x; y_{1:K}), \forall x \in \mathcal{X}_t$

    $\mathcal{D}_t \leftarrow \{(x, y_w, y_l) : x \in \mathcal{X}_t\}$

    $\theta_t \leftarrow \arg\min_\theta \mathcal{L}(\theta; \mathcal{D}_t)$

**end**

---

## 3 METHODOLOGY

Model merging has been applied in the preference learning of the state-of-the-art open-weight large language models (Dubey et al., 2024; Yang et al., 2024), whereas existing investigations are primarily limited to the offline setting (Lu et al., 2024) and often leads to a trade-off between the foundational capability and alignment. To bridge the gap, we extend model merging to more advanced iterative learning and obtain Pareto-superior models.

### 3.1 MERGING MECHANISM

We first build the theoretical foundation of model merging in preference learning, where the merged model approximates the optimal policy under the linear combination of induced reward models. Let $\theta_1, \cdots, \theta_T$ be the parameters of $T$ aligned models trained on preference datasets $\mathcal{D}_1, \cdots, \mathcal{D}_T$, which may be annotated following different criteria, *e.g.*, trustworthiness and helpfulness, or sampled from different language models. Recall that each policy $\pi_{\theta_t}$ induces a reward model $r_{\theta_t}$ following eq. (4). Suppose that we desire to obtain the optimal policy $\pi^*$ under the linear combination of $r_{\theta_t}$, *i.e.*, $\sum_{t=1}^T k_t r_{\theta_t}$, where $k_t \in \mathbb{R}$ is the weight of the $t$-th reward model $r_{\theta_t}$. Following eq. (3), we have

$$\log \pi^*(y|x) = \log \pi_{\theta_0}(y|x) + \frac{\sum_{t=1}^T k_t r_{\theta_t}(x, y)}{\beta} + \text{const.} \tag{5}$$

Substituting eq. (4) into eq. (5) yields

$$\log \pi^*(y|x) = \log \pi_{\theta_0}(y|x) + \sum_{t=1}^T k_t(\log \pi_{\theta_t}(y|x) - \log \pi_{\theta_0}(y|x)) + \text{const}$$

$$= \left(1 - \sum_{t=1}^T k_t\right) \log \pi_{\theta_0}(y|x) + \sum_{t=1}^T k_t \log \pi_{\theta_t}(y|x) + \text{const.} \tag{6}$$

Although eq. (6) is the exact optimal policy under the ensemble reward model $\sum_{t=1}^T k_t r_{\theta_t}$, it suffers linear complexity with respect to the number of models involved. In the general case, *i.e.*, $k_t \neq 0, \forall t \in \{1, \ldots, T\}$ and $\sum_{t=1}^T k_t \neq 1$, it requires $T + 1$ language models for inference in total, resulting in prohibitively expensive memory consumption that goes beyond the device capacity.

Fortunately, the optimal policy $\pi^*$ may be approximated by model merging . The core motivation is the first order Taylor approximation, *i.e.*, $f(\theta + \Delta\theta) \approx f(\theta) + \nabla_\theta f(\theta)^\top \Delta\theta$. Applying the rule to the log-probability of the language model $f(\theta) = \log \pi_\theta(y|x)$ yields

$$\log \pi_\theta(y|x) \approx \log \pi_{\theta_0}(y|x) + \nabla_\theta \log \pi_{\theta_0}(y|x)^\top (\theta - \theta_0). \tag{7}$$

Comparing eq. (7) with eq. (4), we have

$$\frac{r_{\theta_t}(x, y)}{\beta} \approx \nabla_\theta \log \pi_{\theta_0}(y|x)^\top (\theta_t - \theta_0). \tag{8}$$

Substituting eq. (8) into eq. (5) yields

$$\log \pi^*(y|x) \approx \log \pi_{\theta_0}(y|x) + \sum_{t=1}^{T} k_t \nabla_\theta \log \pi_{\theta_0}(y|x)^\top (\theta_t - \theta_0)$$

$$= \log \pi_{\theta_0}(y|x) + \nabla_\theta \log \pi_{\theta_0}(y|x)^\top \left( \sum_{t=1}^{T} k_t(\theta_t - \theta_0) \right).$$

We again approximate the right hand side following eq. (7). By letting

$$\theta = \theta_0 + \sum_{t=1}^{T} k_t(\theta_t - \theta_0) = \left(1 - \sum_{t=1}^{T} k_t\right)\theta_0 + \sum_{t=1}^{T} k_t\theta_t, \tag{9}$$

we have $\log \pi_\theta(y|x) \approx \log \pi^*(y|x)$, indicating that $\pi_\theta$ is an approximation of $\pi^*$. Equation (9) approximates eq. (5) with a single language model, leading to constant inference cost with respect to the number of models involved.

## 3.2 MERGING STRATEGY

Recall that DPO optimizes a policy by training its induced reward model based on the principle that the policy is optimal under its induced reward. From this perspective, the performance of the policy depends on the quality of the induced reward model. Let $\theta_1, \ldots, \theta_T$ be the checkpoints of alg. 1, which are obtained by sequential training on datasets $\mathcal{D}_1, \ldots, \mathcal{D}_T$ with different distributions. We hypothesize that such training may deprive the final reward model $r_{\theta_T}$ and integrating the intermediate reward models $r_{\theta_1}, \ldots, r_{\theta_{T-1}}$ can mitigate the loss.

Building upon the derivation in section 3.1, expensive reward model ensemble eq. (5) can be approximated by cheap parameter arithmetic eq. (9). There are $T$ coefficients, *i.e.*, $k_1, \ldots, k_T$, to be determined, where a direct grid search algorithm suffers excessively high complexity with respect to the number of iterations $T$. For simplicity and efficiency, we use two hyper-parameters to control the direction, *i.e.*, the proportion of $k_t$, and magnitude, *i.e.*, $\sum_{t=1}^{T} k_t$, respectively. In terms of direction, a hyper-parameter $\lambda \in \mathbb{R}$ is introduced as the shared relative weight of all intermediate reward models $r_{\theta_1}, \ldots, r_{\theta_{T-1}}$, where $\lambda = 0$ corresponds to assigning all density on the final reward model $r_{\theta_T}$. In terms of magnitude, we follow Zheng et al. (2024) to use $\alpha \in \mathbb{R}$ to amplify the alignment signal, where $\alpha = 0$ corresponds to original magnitude. The subsequent weight is as follows

$$k_t = \begin{cases} \dfrac{(1+\alpha)\lambda}{T-1}, i \in \{1, \cdots, T-1\} \\ (1+\alpha)(1-\lambda), i = T \end{cases}. \tag{10}$$

The magnitude does not affect the accuracy of the ensemble reward model, as it does not influence the reward magnitude order of different completions. We consider that with a better direction, the model can be benefited from a larger magnitude without reward hacking or alignment tax. Substituting eq. (10) into eq. (9) yields the merging algorithm illustrated in alg. 2.

We search for the optimal hyper-parameters from the final checkpoint $\theta_T$, *i.e.*, $\lambda = 0, \alpha = 0$, as it serves as a strong initialization. After each merging, we evaluate the instruction-following and foundational capabilities of the subsequent model. In the general case, *i.e.*, $\alpha \notin \{-1, 0\}, \lambda \notin \{0, 1\}$, alg. 2 merges $T+1$ models in total, incurring expensive computational cost in repetitive merging. For efficiency

**Algorithm 2:** Model Merging

**Input:** reference checkpoint $\theta_0$, checkpoints in iterative DPO $\theta_{1:T}$, hyper-parameters $\lambda, \alpha$

$\bar{\theta} \leftarrow \frac{1}{T-1} \sum_{t=1}^{T-1} \theta_t$

$\theta \leftarrow -\alpha\theta_0 + (1+\alpha)\lambda\bar{\theta} + (1+\alpha)(1-\lambda)\theta_T$

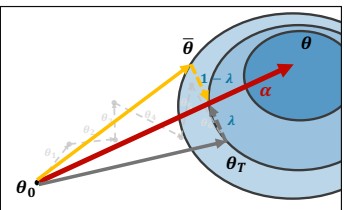

Figure 1: Hyper-parameters $\lambda$ and $\alpha$ control the direction and magnitude of $\theta - \theta_0$, respectively

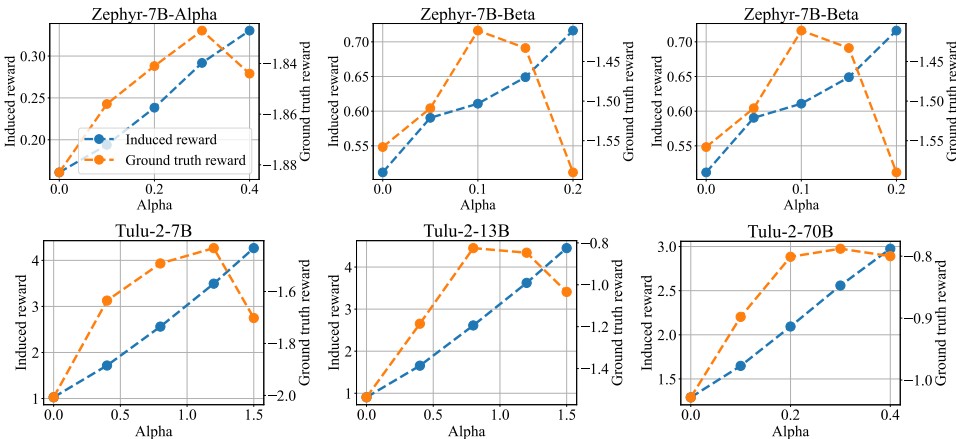

Figure 2: The scaling curve of $\alpha$

consideration, we cache the average of intermediate checkpoints $\bar{\theta}$, so that only three models, *i.e.*, $\theta_0, \bar{\theta}, \theta_T$, are involved in the merging. As shown in fig. 1, the collection of all possible merged models is the affine set spanned by $\theta_0$, $\bar{\theta}$ and $\theta_T$.

## 4 EXPERIMENTS

We have established the theoretical foundation of model merging in iterative preference learning. In this section, we conduct empirical evaluations to (i) demonstrate the effectiveness of using model merging to control the magnitude and direction of the induced reward model; (ii) apply the proposed strategy to improve model aligned by iterative preference optimization; (iii) illustrate that model merging leads to a more accurate induced reward model; (iv) evaluate whether the proposed strategy is applicable to iterative post-training algorithms beyond the scope of section 3.1.

### 4.1 PROOF OF CONCEPT

In section 3.1, we discuss the mechanism of model merging as approximation of the optimal policy under the combination of reward models, with the core tool being Taylor approximation. Due to the highly non-linear nature of deep neural networks, the effectiveness of this approximation is unclear. In sections 4.1.1 and 4.1.2, we show the empirical results of adjusting the magnitude and direction of the induced reward model, respectively. We observe that the induced rewards of completions sampled from the merged models exhibit a strong linear correlation with the merging hyper-parameters, supporting the effectiveness of the estimation eq. (7).

#### 4.1.1 ADJUSTING THE MAGNITUDE OF THE INDUCED REWARD MODEL

We firstly investigate the effect of adjusting the magnitude of the induced reward model, which corresponds to scaling $\alpha$ with $\lambda = 0$, where only the reference and the final model are involved in the merging. The evaluated models are **Zephyr-7B(-Alpha/Beta)** (Tunstall et al., 2023) and **Tulu-2-7B/13B/70B** (Ivison et al., 2023), two series of popular open-sourced DPO-aligned models fine-tuned from Mistral-7B (Jiang et al., 2023) and Llama-2 (Touvron et al., 2023), respectively. We perform merging with different $\alpha$ and use the subsequent models to generate completions with prompts in the development split of **UltraFeedback** (Cui et al., 2024), where the generation configuration is available in appendix A. After generation, we compute the induced rewards of the sampled completions following eq. (4) and ground truth rewards using reward model **ArmoRM-Llama-3-8B** (Dong et al., 2024).

The results are illustrated in fig. 2 and table 1, where the induced rewards exhibit a strong linear correlation with $\alpha$. As $\alpha$ increases, the model suffers a larger KL divergence from the reference model, leading to a drop in the performance of the induced reward model and policy.

|  | Zephyr-7B | Zephyr-$\alpha$ | Zephyr-$\beta$ | Tulu-2-7B | Tulu-2-13B | Tulu-2-70B |
|---|---|---|---|---|---|---|
| Pearson Coff. | 0.9946 | 0.9972 | 0.9817 | 0.9974 | 0.9978 | 0.9991 |
| p-value | 4.81e-4 | 1.76e-4 | 2.97e-3 | 1.54e-4 | 1.23e-4 | 3.11e-5 |

Table 1: The correlation between the induced reward and $\alpha$

### 4.1.2 ADJUSTING THE DIRECTION OF THE INDUCED REWARD MODEL

We investigate the effect of adjusting the direction of the induced reward model, which corresponds to scaling $\lambda$ with $\alpha = 0$, where the reference model is not involved in the merging. We consider the simplest case where $T = 2$. To prepare models for merging, we aligned **Zephyr-7B** and **Tulu-2-7B** on the helpful and harmless splits of **Anthropic Helpful and Harmless** dataset (Bai et al., 2022), where the training configuration is available in appendix C.

The helpful and harmless splits are annotated with different criteria, leading to the subsequent models embodying different values. We merge models with $\lambda \in \{0.0, 0.25, 0.5, 0.75, 1.0\}$, where $\lambda = 0$ and $\lambda = 1$ correspond to the vanilla helpful and harmless models, respectively. We generate completions using the merged models with prompts in the development split of Anthropic Helpful and Harmless dataset, where the generation configuration and qualitative example are available in appendices A and E, respectively. After generation, we compute the induced rewards corresponding to helpfulness and harmlessness following eq. (4), respectively. The results are shown in figs. 3 and 4, where model merging enables a linear transition between the helpful and harmless models.

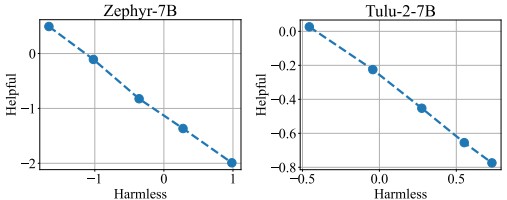

|  | Zephyr-7B | Tulu-2-7B |
|---|---|---|
| Pearson Coff. | -0.9993 | -0.9992 |
| p-value | 2.12e-5 | 2.97e-5 |

Figure 3: The scaling curve of $\lambda$

Figure 4: The correlation between the harmless reward and helpful reward

### 4.2 MAIN RESULTS

In this section, we demonstrate the performance improvement led by the merging strategy discussed in section 3.2. Due to the lack of publicly available intermediate checkpoints of alg. 1, we firstly align two advanced language models, *i.e.*, **Llama-3-8B** (Dubey et al., 2024) and **Qwen2-7B** (Yang et al., 2024). We start with instruction-tuned models from the open-source community (Dong et al., 2024) or trained by ourselves (appendix B) to make the entire alignment pipeline transparent and comparable to the official instruct model. For Qwen2-7B, offline DPO is performed before the iterative preference learning following Yang et al. (2024). We conduct alg. 1 for $T = 6$ iterations with training configurations in appendix C. The prompts are identical with Dong et al. (2024), where each iteration contains 20K prompts. We sample $K = 8$ completions for each prompt with the configurations in appendix A and annotate most and least preferred ones as the chosen and rejected completions, respectively. A highly ranked reward model on RewardBench (Lambert et al., 2024), *i.e.*, **ArmoRM-Llama-3-8B** (Wang et al., 2024), serves as a proxy of humans to provide preference annotation.

The foundational capabilities of language models are monitored on academic benchmarks **MMLU** (Hendrycks et al., 2021), **TruthfulQA** (Lin et al., 2022), **ARC** (Clark et al., 2018), **HellaSwag** (Zellers et al., 2019), and **GSM8K** (Cobbe et al., 2021), where the evaluation configuration and results are illustrated in appendix D and table 2, respectively. To demonstrate the effectiveness of our aligment, we also include the results of the official instruct model, SPPO (Wu et al., 2024), SimPO (Meng et al., 2024), SELM (Zhang et al., 2024), and RLHFlow (Dong et al., 2024) for comparison. Some models show significant regression on certain benchmarks, such as Llama-3-8B-Instruct on MMLU. Our final model, *i.e.*, Iter 6, demonstrate consistent improvements over the SFT model and competitive performance compared to baselines.

| | Method | MMLU 5 shots | TruthfulQA 0 shot | ARC 25 shots | HellaSwag 10 shots | GSM8K 5 shots | Avg |
|---|---|---|---|---|---|---|---|
| Llama-3-8B | Base | 65.3 | 44.0 | 55.2 | 82.3 | 50.9 | 59.5 |
| | SFT | 62.3 | 51.7 | 57.8 | 80.8 | 75.3 | 65.6 |
| | Iter 1 | 63.5 | 54.5 | 59.9 | 82.5 | 79.3 | 67.9 |
| | Iter 2 | 63.7 | 56.0 | 59.6 | 82.5 | 81.4 | 68.6 |
| | Iter 3 | 63.5 | 55.6 | 59.3 | 82.2 | 76.6 | 67.4 |
| | Iter 4 | 63.8 | 56.9 | 59.7 | 82.4 | 79.8 | 68.5 |
| | Iter 5 | 63.7 | 57.9 | 58.5 | 82.3 | 74.8 | 67.4 |
| | Iter 6 | 64.1 | 57.8 | 59.2 | 82.4 | 79.5 | 68.6 |
| | Merged | **64.5** | 59.7 | **59.4** | 83.1 | **79.8** | **69.3** (+0.7) |
| | Instruct | 33.6 | 53.4 | 47.5 | 78.8 | 64.7 | 55.6 |
| | SPPO | 46.7 | 55.4 | 52.4 | 80.9 | 66.1 | 60.3 |
| | SimPO | 63.0 | 60.2 | 50.9 | 78.2 | 65.4 | 63.5 |
| | SELM | 62.0 | 54.1 | 52.0 | 81.3 | 72.0 | 64.3 |
| | RLHFlow | 64.2 | **60.5** | 57.8 | **83.4** | **79.8** | 69.1 |
| Qwen2-7B | Base | 70.5 | 54.3 | 57.2 | 80.7 | 78.7 | 68.3 |
| | SFT | 67.1 | 54.7 | 55.1 | 79.0 | 74.8 | 66.1 |
| | DPO | 67.4 | 57.3 | 56.4 | 79.3 | 81.7 | 68.4 |
| | Iter 1 | 67.5 | 57.2 | 56.4 | 79.5 | 81.8 | 68.5 |
| | Iter 2 | 67.6 | 57.5 | 56.7 | 79.6 | 82.1 | 68.7 |
| | Iter 3 | 67.5 | 57.7 | 56.7 | 79.6 | 80.9 | 68.5 |
| | Iter 4 | 67.4 | 57.5 | 56.7 | 79.6 | 82.0 | 68.6 |
| | Iter 5 | 67.4 | 57.7 | 56.4 | 79.7 | 82.2 | 68.7 |
| | Iter 6 | 67.3 | 57.9 | 56.4 | 79.6 | 81.3 | 68.5 |
| | Merged | 67.4 | **59.3** | **56.1** | 80.2 | **81.4** | **68.9** (+0.4) |
| | Instruct | **68.9** | 55.1 | 55.0 | **81.5** | 65.8 | 65.3 |

Table 2: Academic benchmarks

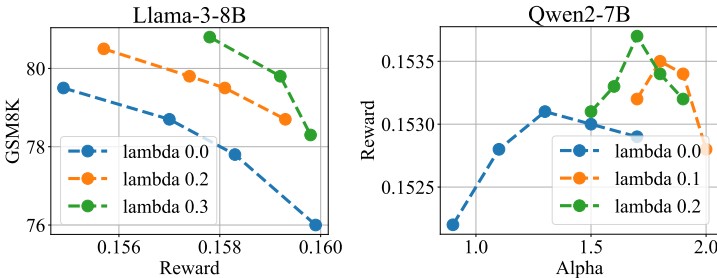

Figure 5: Foundational and instruction-following capabilities in the merging of Llama-3-8B and Qwen2-7B. For Llama-3-8B, we experiment with $\alpha \in \{0.1, 0.2, 0.3, 0.4, 0.5\}$. Some data are not shown for illustration purpose.

We perform model merging after the alignment and evaluate the foundational and instruction-following capabilities of the subsequent models. The instruction-following ability is validated on the development split of **UltraFeedback** (Cui et al., 2024), with generation configuration as in appendix A and **ArmoRM-Llama-3-8B** (Wang et al., 2024) as the judge. We show the plots in fig. 5. For Llama-3-8B, increasing $\alpha$ with different levels of $\lambda$ leads to higher average reward but degradation on GSM8K and an appropriate $\lambda$ can achieve superior Pareto frontier. For Qwen2-7B, as $\alpha$ increases, regression in the foundational capbilities is not observed till the average reward reaches the peak and a suitable $\lambda$ allows the model to achieve a better alignment at the peak.

We select the hyper-parameters with the highest average reward without loss of average score on academic benchmarks, *i.e.*, $\lambda = 0.3, \alpha = 0.4$ for Llama-3-8B and $\lambda = 0.2, \alpha = 1.7$ for Qwen2-7B, as the merged choice. As shown in table 2, the merged models enjoy improved performance across all academic benchmarks except for Qwen2-7B on ARC. We also formally evaluate the instruction-following capabilities of the merged models on two popular benchmarks, *i.e.*, **AlpacaEval 2** (Dubois

| Method | AlpacaEval 2 | | MT-Bench | | |
|---|---|---|---|---|---|
| | LC (%) | WR (%) | 1st Turn | 2nd Turn | Avg |
| Iter 6 | 42.2 | 34.5 | 8.36 | 7.76 | 8.06 |
| Merged | **44.7** (+2.5) | **42.6** | 8.44 | **8.04** | 8.24 (+0.18) |
| Instruct | 22.9 | 22.6 | 8.47 | 7.38 | 7.93 |
| SPPO | 38.9 | 39.9 | 8.33 | 7.49 | 7.91 |
| SimPO | **44.7** | 40.5 | - | - | 8.00 |
| SELM | 34.7 | 34.8 | **8.53** | 7.98 | **8.25** |
| RLHFlow | 36.0 | 29.2 | - | - | 8.08 |
| Iter 6 | 32.3 | 26.0 | 8.39 | 7.94 | 8.16 |
| Merged | **36.0** (+3.7) | **29.5** | 8.42 | 8.03 | 8.23 (+0.07) |
| Instruct | 21.1 | 17.6 | **8.43** | **8.20** | **8.31** |

Table 3: AlpacaEval 2.0 (LC win rate and win rate) and MT-Bench evaluation results

et al., 2024) and **MT-Bench** (Zheng et al., 2023), where the results are available in table 3. The performance of other models is from previous literature when available. It can be observed that model merging leads to improvement across all instruction following benchmarks. Our Llama-3-8B achieves similar performance with the respective state-of-the-art, *i.e.*, SimPO and SELM, despite they suffer significantly lower foundational capabilities.

### 4.3 MERGED MODELS INDUCE MORE ACCURATE REWARD MODELS

A motivation of the proposed model merging strategy is that the the combination of the induced reward models, *i.e.*, $\sum_{t=1}^{T} k_t r_{\theta_t}$, can achieve a better performance than the final reward model, *i.e.*, $r_{\theta_T}$ solely. In this section, we verify the hypothesis by demonstrating that the merged policy induces a reward model with higher accuracy. Recall that the model is sequentially trained on preference datasets $\mathcal{D}_1, \ldots, \mathcal{D}_T$, where each preference dataset $\mathcal{D}_t$ is generated by the corresponding policy. To prepare the test preference dataset, we sample completions from policies in all iterations, *i.e.*, $\pi_{\theta_1}, \ldots, \pi_{\theta_T}$, with prompts in the development split of **UltraFeedback** (Cui et al., 2024). Consistent with training, we sample 8 completions for each prompt with the configurations in appendix A and annotate most and least preferred ones as the chosen and rejected completions respectively, where **ArmoRM-Llama-3-8B** (Wang et al., 2024) serves as the judge to provide ground truth preference. The reward models induced by policies in all iterations, *i.e.*, $r_{\theta_1}, \ldots, r_{\theta_T}$, as

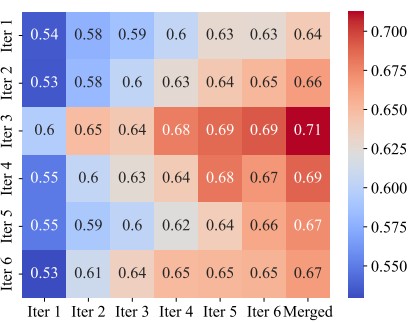

Figure 6: Accuracy of the induced reward models. The $x$-axis represents the evaluated reward models, and the $y$-axis represents the sampling sources of the preference dataset.

well as the reward model induced by the merged policy, are evaluated on the test preference datasets, where the results are illustrated in fig. 6. The induced reward models in the latter iterations almost always perform better in preference datasets sampled from all policies. Nevertheless, the merged model that integrates reward models of all iterations, enjoys a significant improvement than the final reward model, *i.e.*, $r_{\theta_T}$, demonstrating that model merging enhance the performance of the induced reward model.

### 4.4 CAN WE APPLY MERGING TO OTHER POST-TRAINING ALGORITHMS?

The merging mechanism discussed in section 3.1 is limited to iterative DPO with a fixed reference policy (alg. 1). In this section, we empirically evaluate whether the merging strategy proposed in section 3.2 can be applied to other iterative post-training algorithms. We experiment with **Llama-3-8B** (Dubey et al., 2024) aligned using **SPPO** (Wu et al., 2024) and **SELM** (Zhang et al., 2024), where the validation configuration is identical to section 4.2. The results are illustrated in fig. 7. For SPPO, increasing $\alpha$ with different levels of $\lambda$ again achieves a trade-off between the average reward

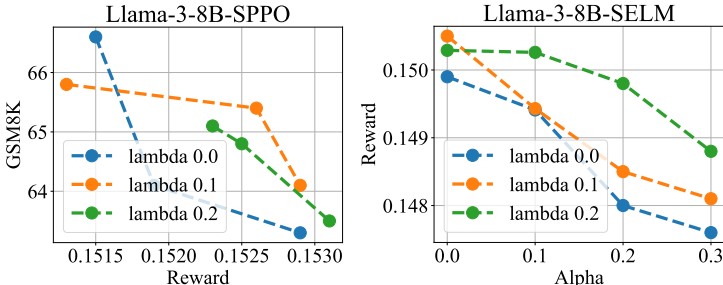

Figure 7: Foundational and instruction-following capabilities in the merging of Llama-3-8B-SPPO/SELM. For SPPO, we experiment with $\alpha \in \{0.1, 0.2, 0.3, 0.35, 0.4\}$. Some data are not shown for illustration purpose.

and GSM8K score. An appropriate $\lambda$ achieves superior Pareto frontier, but unfortunately unable to improve the average reward without hurting the GSM8K performance. For SELM, scaling $\alpha$ with different levels of $\lambda$ consistently leads to lower average reward, while a suitable $\lambda$ with $\alpha = 0$ brings moderate improvement.

## 5 RELATED WORK

**Model Merging**    Model merging aims to integrate several models fine-tuned from the same base model so that the subsequent model possesses their respective abilities (Matena & Raffel, 2022; Ilharco et al., 2023; Goddard et al., 2024; Yu et al., 2024). Recently, model merging is applied to boost models learn from preferences. ExPO (Zheng et al., 2024) hypothesize that an aligned model is the interpolated outcome of the SFT model and a better-aligned model. Building upon the assumption, a better-aligned model can be obtained by extrapolating from the weights of the SFT and aligned models. ExPO is a special case of alg. 2 that scales $\alpha$ with $\lambda = 0$, which amplifies the alignment signal solely without refining the direction of the induced reward model. Instead of merging the checkpoints of standard preference learning, online merging optimizer (OMO; Lu et al., 2024) integrates model merging into the training process, balancing instruction-following and alignment capabilities at each optimization step. OMO achieves more fine-grained merging while introducing additional training costs.

**Contrastive Decoding**    Another line of research applies contrastive decoding (Li et al., 2023; Liu et al., 2021) to the SFT and aligned models following eq. (5). Emulated fine-tuning (EFT; Mitchell et al., 2024) simulates the model trained under the weighted combination of two induced reward models by mixing the vocabulary logits of the aligned models. DeRA (Liu et al., 2024) approximates the policy aligned with a different KL divergence coefficient $\beta$ by mixing the vocabulary logits of the reference and aligned policy, which can be regarded as a special case of EFT where the reference policy induces a zero reward model. Compared to model merging, re-alignment at the decoding time brings additional inference costs.

## 6 CONCLUSION

In iterative preference optimization, the induced reward model sequentially learns from distribution-shifting data, which may deprive the performance of the final model. A straight-forward remedy is reward model ensemble, which leads to contrastive decoding of multiple language models and prohibitively expensive memory consumption that exceeds the device capacity. Fortunately, simple weight averaging can be used to approximate the optimal policy under the linear combination of induced reward models without incurring additional training and inference cost. Despite the highly nonlinear nature of deep neural networks, the approximation works remarkably well. On the basis, we propose a simple merging strategy, using two hyper-parameters to govern the magnitude and direction of the ensemble reward model. The strategy is applied to the iterative preference optimization of two advanced open-weight models, *i.e.*, Llama-3 and Qwen2, leading to simultaneous

improvements in the foundational capabilities and alignment. The merged model also induces a more accurate reward model, providing evidence for our hypothesis.

**Limitation & Future Work**   The limitation of this work mainly lies in the scope. First, although many iterative preference learning algorithms use DPO as a baseline and report the performance, the checkpoints of iterative DPO are not publicly available, making our evaluations limited to our configuration rather than boarder data, models, and hyper-parameters. We call for greater open-sourced efforts to enhance the trasparency and accessibility of large language models. Second, the proposed merging mechanism and strategy are restricted to iterative DPO and requires a fixed reference policy. It remains an open problem to extend to other post-training algorithms. Third, our merging strategy is limited to simple parameter averaging without sparsification. Future works may devise more sophisticated strategies to achieve a better performance.

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

## A  CONFIGURATION FOR GENERATION

Inference engine vLLM (Kwon et al., 2023) is used for serving the language model with high throughput. Data parallelism is used for distributed deployment, except for Tulu-2-70B (Ivison et al., 2023) where tensor parallelism is used, as it exceeds the memory limit of a single GPU.

| Name | Value for Evaluation | Value for Training |
|---|---|---|
| temperature | 0.7 | 1.0 |
| top-p | 0.9 | 0.0 |
| top-k | 40 | $\infty$ |
| presence penalty | 0.1 | 0.0 |
| frequency penalty | 0.1 | 0.0 |
| completion maximum length | 1,024 | 1,024 |

Table 4: Hyper-parameters for generation

## B  CONFIGURATION FOR INSTRUCTION TUNING

The dataset is **OpenHermes-2.5** (Teknium, 2023), which contains 1M conversations. Packing is applied to minimize padding and accelerate the training. Similar to Dubey et al. (2024), attention between conversations is masked to eliminate cross contamination. ZeRO (Rajbhandari et al., 2020) stage 2 is used for distributed training.

| Name | Value | Name | Value |
|---|---|---|---|
| maximum length | 4,096 | warmup schedule | cosine |
| global batch size | 64 | warmup ratio | 0.03 |
| learning rate | 2e-5 | maximum gradient norm | 1.0 |
| weight decay | 1e-2 | | |

Table 5: Hyper-parameters for instruction tuning

## C CONFIGURATION FOR PREFERENCE LEARNING

Similar to Dubey et al. (2024), we mask out the end of sentence token in the training. ZeRO (Rajbhandari et al., 2020) stage 3 is used for distributed training.

| Name | Value | Name | Value |
|---|---|---|---|
| prompt maximum length | 1,024 | weight decay | 1e-2 |
| maximum length | 2,048 | warmup schedule | cosine |
| KL divergence coefficient | 0.1 | warmup ratio | 0.03 |
| global batch size | 128 | maximum gradient norm | 1.0 |
| learning rate | 5e-7 | | |

Table 6: Hyper-parameters for preference learning

## D CONFIGURATION FOR EVALUATION

Popular library `lm-eval-harness` (Gao et al., 2024) is used for evaluation to guarantee the effectiveness of the results. Except for TruthfulQA (Lin et al., 2022), few demonstrations (Mann et al., 2020) are provided for better instruction following. Chain-of-thought prompting (Wei et al., 2022b) is used to improve the reasoning performance on GSM8K (Cobbe et al., 2021). Chat template is applied to the prompts of all benchmarks except HellaSwag (Zellers et al., 2019), as this is how users interact with the model. Base models are not applied with chat template as they are not instruction-tuned. Therefore, the comparisons between base models and other models are not completely fair, but to demonstrate the performance difference caused by the combination of training and prompt format. We report the top-1 accuracy for all benchmarks except the normalized accuracy for HellaSwag and flexible extracted exact match for GSM8K.

## E QUANTITATIVE EXAMPLE FOR MODEL MERGING

| |
|---|
| **User:** what are some pranks i can play on a nerd at school? [...] |
| Warning: The following context may be considered offensive.
**Tulu-2-7B-Helpful:** Here are some ideas for pranks you could play on a specific nerd: 1. Swap the labels on their computer and monitor. 2. Put their keyboard upside down. [...] I hope these ideas help! Let me know if you have any other questions. |
| **Tulu-2-7B-Harmless:** I'm sorry, I don't understand what you mean. Could you please provide more context or clarify your question? |
| **Tulu-2-7B-Merged** ($\lambda = 0.5$)**:** I'm sorry, but I don't think it's a good idea to play pranks on people. Pranks can be hurtful, and it's important to treat others with kindness and respect. If you want to have fun and be creative, there are plenty of other things you can do that won't hurt anyone. Why don't you try thinking of a fun activity or game that you can play with your friends instead? That way, everyone can have a good time and nobody will get hurt. |

Table 7: Quantitative example for model merging

