# OpenReview forum: "Language Model Merging in Iterative Preference Learning"
_ICLR.cc/2025/Conference — ICLR 2025 Conference Withdrawn Submission_

### Official Review · Reviewer_YH2y · 2024-10-29

**Soundness:** 3
**Presentation:** 3
**Contribution:** 2
**Rating:** 5
**Confidence:** 3

**Summary:**

This paper proposes a method to merge the checkpoints derived from iterative preference optimization. The authors provide both rigorous theoretical proof and experiment results to demonstrate the merged models can approximate ensembled rewards. Experiments are conducted on two open source LLMs and five benchmarks, to prove the method can produce models with better foundational capabilities.

**Strengths:**

1. The method is based on solid theoretical foundations. Meanwhile, proof-of-concept experiments are conducted to validate the effectiveness of the approximation in the theoretical derivation, which makes the proposed approach technically sound.
2. The experimental design directly addresses concerns related to the theoretical approximation and effectively demonstrates the effectiveness of the merging strategy.
3. The writing is clear and easy to follow. The presentation of the method and the flow of the experimental results are smooth and well-structured. However, the abstract and introduction need improvement, see weaknesses section for details.

**Weaknesses:**

1. The comparison with previous baselines is insufficient. Results are only provided for dpo training baselines on Llama-8B, which weakens the authors' claims of effectiveness.
2. The main results are not compelling. The merged model does not demonstrate a clear performance gain compared to the individual checkpoints used for merging.
3. ExPO is highly relevant to the proposed method, simply mentioning that ExPO is a special case of Eq(2) is not enough. More experimental comparisons between ExPO's model merging method and the proposed method are needed, both on Qwen2 and Llama3.
4. The abstract is overly brief and does not adequately focus on the proposed method or the experimental results. Similarly, the introduction provides only a general overview of iterative preference learning, with little attention to the motivation behind the proposed approach. The only mention of motivation is on Line 56: "such training may deprive the performance of the final reward model and policy." However, this is presented as a hypothesis, without any preliminary experimental evidence to support it. This severely limits the overall contribution of the paper.
4. The figure captions lack detail and clarity. For instance, in Figure 2, "the scaling curve of $\alpha$" could benefit from a more thorough explanation, particularly regarding the y-axis and its significance. Figure 4 suffers from the same issue.
5. Captions for Figures 5 and 7 should include a summary of the key takeaway; otherwise, their purpose is unclear. In addition, the statement that "some data are not shown for illustration purposes" is vague. The authors should explain what data is omitted and why its exclusion does not affect the main point they aim to convey.
6. The abstract claims the proposed method can generate Pareto-superior models, but this is not convincingly supported by the experimental results. It's unclear to what method the Pareto frontier is being compared, and in which aspects the comparison is being made.

**Questions:**

See weaknesses.

---

### Official Review · Reviewer_nYrN · 2024-10-30

**Soundness:** 3
**Presentation:** 3
**Contribution:** 2
**Rating:** 5
**Confidence:** 4

**Summary:**

This paper, starting from the perspective of the first-order Taylor expansion, proposes a weight averaging method to ensemble reward models. It demonstrates the effectiveness of this approach in aligning LLMs through iterative preference optimization.

**Strengths:**

- The paper is well-written and easy to follow.
- The motivation is clear and straightforward.
- By performing model weighting after iterative alignment of LLMs, the authors achieve the effect of blending multiple policy distributions. Compared to ensembling models during inference, this approach significantly reduces inference costs.

**Weaknesses:**

- The terms "reward modelling" (L73) and "reward modeling" need to be consistent throughout the paper.
- The approach uses equal weights for all models except the final one, which seems somewhat trivial. Since $T_i$ is fine-tuned from $T_{i-1}$, I am curious about the impact of other weighting strategy.
- In Table 2, while the merged model shows improvement compared to Iter 6, the performance gains relative to the SoTA models are inconsistent.

**Questions:**

- Figure 1: The figure indicates that $\alpha$ controls the magnitude of $\theta - \theta_0$, but I believe it should control the magnitude of the interpolation between $\bar{\theta}$ and $\theta_T$, minus $\theta_0$. Could you clarify this further?

- I am curious about the source of the SPPO results presented in the paper. The SPPO paper reports that Llama-3-8B-SPPO Iter3 achieves an average score of 70.29 on the OpenLLM leaderboard, which is higher than the results the authors presented. Are there any differences in settings that might explain this?

---

### Official Review · Reviewer_34HX · 2024-11-03

**Soundness:** 2
**Presentation:** 3
**Contribution:** 2
**Rating:** 5
**Confidence:** 4

**Summary:**

In the context of iterative preference optimization, model ensemble is an effective method but incurs high inference costs. This paper introduces a model ensemble strategy that approximates model ensemble via weights merging between the reference policy and iterative DPO checkpoints. Experiments with Llama-3 and Qwen2 on several benchmarks demonstrate its effectiveness in improving the model's foundational capabilities and alignment performance.

**Strengths:**

1. The proposed method is easy to implement, and it shows improvements in the setting of iterative preference learning.

2. The experiments cover several representative model families and examine enough representative tasks.

**Weaknesses:**

1. My major concern comes from the core motivation introduced in lines 154 - 161, where the Taylor approximation is applied. However, the key assumption for the first-order Taylor series expansion to be a good approximation is that $\Delta \theta$ should be sufficiently small relative to $\theta$, so the higher-order terms in the Taylor expansion can be safely ignored. The authors did not illustrate whether this important assumption holds.

2. For the main results reported in Table 2, it needs to discuss the training costs of the proposed method and the baselines to ensure a fair comparison. It is hard to tell whether the performance gain comes from the training overhead based on the limited implementation details as described in the current version.

3. While discussed in related works, the model merging methods are not compared in experiments. It will provide a more comprehensive understanding of the proposed method by including other model merging methods in experiments.

4. Missing reference.
[1] WARM: On the Benefits of Weight Averaged Reward Models.

**Questions:**

1. Can you justify whether the first-order Taylor approximation is reasonable or show higher-order Taylor approximation results?

2. Can you present more implementation details to show whether the comparison to baselines is fair?

---

### Official Review · Reviewer_ABU3 · 2024-11-04

**Soundness:** 2
**Presentation:** 3
**Contribution:** 2
**Rating:** 5
**Confidence:** 4

**Summary:**

The paper presents a merging strategy for language models in the context of iterative preference learning. The authors propose a technique that approximates ensemble methods through parameter averaging of previous checkpoints, which enhances alignment and foundational capabilities without incurring extra training or inference costs.

**Strengths:**

1. The paper is well-written with clear logic, making it easy to follow and understand.

---
2. The experiments appear effective, demonstrating the proposed method’s benefits.

---
3. The PRELIMINARY section is introduced clearly, providing a solid foundation and context for the methodology that follows.

---
4. The focus on iterative preference optimization rather than offline preference optimization is practical.

**Weaknesses:**

1. The use of a first-order Taylor approximation assumes that $\log \pi_\theta(y \mid x)$ is approximately linear around $\theta_0$. However, this simplification overlooks the highly non-linear nature of deep language models due to complex activation functions like softmax. This means that higher-order terms, which can have significant effects, are ignored, potentially leading to inaccurate approximations. Additionally, the assumption that $\theta_t$ is close to $\theta_0$ may not hold in practice, especially when parameter differences are large, further reducing the reliability of the approximation.

---
2. The statement: "We hypothesize that such training may deprive the performance of the final reward model and policy," lacks references or empirical evidence. This makes it difficult to confirm that the identified issue exists or that the proposed solution effectively addresses it, thus weakening the rationale behind the study.

---
3. The paper does not discuss related work that shares conceptual similarities, such as [1]. While the primary difference is that the merged checkpoints in the proposed method are derived intuitively rather than offline.

[1] Rewarded Soups: Towards Pareto-optimal Alignment by Interpolating Weights Fine-Tuned on Diverse Rewards


---
4. Although the authors emphasize their scenario in an iterative setting rather than offline, the technique of using linear interpolation of parameter is fairly standard. The method seems to have been applied specifically to the iterative scenario without significant innovation in the underlying approach, which reduces the overall novelty of the contribution.


---
5. The paper does not address potential failures associated with the parameter-merging paradigm, as discussed in [2]. These known failures are likely applicable to your proposed method as well, and a discussion on this would contribute to a more balanced evaluation of the approach's limitations.

[2] Decoding-Time Language Model Alignment with Multiple Objectives

**Questions:**

I'm wondering if the potential failures associated with the parameter-merging paradigm, as discussed in [2], are present in your method when applied iteratively. If so, how do you plan to address or mitigate these limitations?

[2] Decoding-Time Language Model Alignment with Multiple Objectives

---

### Official Review · Reviewer_gNiY · 2024-11-04

**Soundness:** 2
**Presentation:** 3
**Contribution:** 1
**Rating:** 3
**Confidence:** 4

**Summary:**

The paper addresses the issue of distributional inconsistencies in Iterative Preference Learning and proposes a model merging approach to achieve Pareto-optimal outcomes, supported by a theoretical analysis grounded in Preference Learning principles.
Two key hyperparameters are introduced to control the magnitude and direction of the reward model ensemble, enhancing the effectiveness of the model merging strategy.
 It would be beneficial to clearly summarize and emphasize the core contributions of the paper.
The central novelty of this work is difficult to grasp. On one hand, model merging techniques have already been explored for (iterative) alignment in prior research. On the other hand, the paper claims that model merging results in a more accurate reward model, which is then utilized for alignment. If I have misunderstood this, I would appreciate clarification.

**Strengths:**

- The paper is well-written, making it relatively accessible and engaging for readers.
 - The theoretical analysis provided for preference learning is robust and well-supported.

**Weaknesses:**

- The innovation of using model merging in the context of iterative preference learning appears limited, as similar techniques have been applied in the LLaMA 3 series [1]. Combining model merging with SimPO to create more powerful models seems like a more compelling direction to explore.
 - The primary experimental results focus on the merging of alignment models rather than reward models, which weakens the evidence supporting the paper’s claims about its effectiveness for reward modeling. The overall objective of the paper needs to be articulated more clearly.
- The link between the claim of achieving more accurate reward models and the performance improvements in iterative optimization is insufficiently developed. Given that the ArmoRM reward model is used offline in iterative alignment, it is essential to conduct experiments using DPO-based reward models for iterative learning to validate the approach [2].


[1] The LLaMA 3 Herd of Models

[2] Lambert, Nathan, et al. “Rewardbench: Evaluating reward models for language modeling.” _arXiv preprint arXiv:2403.13787_ (2024).

**Questions:**

- What is the rationale behind using “Figure 4,” which seems to be more appropriately presented as “Table 2”?
- Does the alignment with six iterations introduce significantly higher training costs compared to more advanced baselines, such as SimPO?
- The motivation for introducing more accurate reward models in Section 4.3 is unclear. Specifically, what does it mean to obtain a more accurate reward model based on the DPO framework? Additionally, since ArmoRM is utilized to select preferences, have you explored using DPO as a reward model (RM) to select preferences during the preference learning process? Clarifying this would greatly improve the understanding of your methodology.

---

### Note · Authors · 2024-11-23

I have read and agree with the venue's withdrawal policy on behalf of myself and my co-authors.